# Comparison of BMIPP-SPECT/CT to ^18^FDG-PET/CT for Imaging Brown or Browning Fat in a Preclinical Model

**DOI:** 10.3390/ijms23094880

**Published:** 2022-04-28

**Authors:** Joseph A. Frankl, Yu An, Amber Sherwood, Guiyang Hao, Feng-Yun Huang, Pawan Thapa, Deborah J. Clegg, Xiankai Sun, Philipp E. Scherer, Orhan K. Öz

**Affiliations:** 1Department of Radiology, University of Texas Southwestern Medical Center, 5323 Harry Hines Blvd, Dallas, TX 75390, USA; joseph.frankl@utsouthwestern.edu (J.A.F.); amber.sherwood@utsouthwestern.edu (A.S.); guiyang.hao@utsouthwestern.edu (G.H.); pawan.thapa@utsouthwestern.edu (P.T.); xiankai.sun@utsouthwestern.edu (X.S.); 2Department of Anesthesiology, McGovern Medical School, University of Texas Health Science Center at Houston, Houston, TX 77030, USA; yu.an@uth.tmc.edu; 3Department of Medical Imaging and Radiological Sciences, Central Taiwan University of Science and Technology, No. 666, Buzih Road, Beitun District, Taichung City 406053, Taiwan; fyhuang@ctust.edu.tw; 4Department of Internal Medicine, Texas Tech Health Sciences Center, 5001 El Paso Dr, El Paso, TX 79905, USA; dclegg@ttuhsc.edu; 5Touchstone Diabetes Center, Department of Internal Medicine, Southwestern Medical Center, University of Texas, 5323 Harry Hines Blvd, Dallas, TX 75390, USA; philipp.scherer@utsouthwestern.edu

**Keywords:** brown fat, white adipose tissue, BMIPP, FDG, beta-adrenergic receptor, PET/CT, SPECT/CT

## Abstract

Obesity is a leading cause of preventable death and morbidity. To elucidate the mechanisms connecting metabolically active brown adipose tissue (BAT) and metabolic health may provide insights into methods of treatment for obesity-related conditions. ^18^F-fluorodeoxyglucose positron emission tomography/computed tomography (^18^FDG-PET/CT) is traditionally used to image human BAT activity. However, the primary energy source of BAT is derived from intracellular fatty acids and not glucose. Beta-methyl-p-iodophenylpentadecanoic acid (BMIPP) is a fatty acid analogue amenable to in vivo imaging by single photon emission computed tomography/CT (SPECT/CT) when radiolabeled with iodine isotopes. In this study, we compare the use of ^18^FDG-PET/CT and ^125^I-BMIPP-SPECT/CT for fat imaging to ascertain whether BMIPP is a more robust candidate for the non-invasive evaluation of metabolically active adipose depots. Interscapular BAT, inguinal white adipose tissue (iWAT), and gonadal white adipose tissue (gWAT) uptake of ^18^FDG and ^125^I-BMIPP was quantified in mice following treatment with the BAT-stimulating drug CL-316,243 or saline vehicle control. After CL-316,243 treatment, uptake of both radiotracers increased in BAT and iWAT. The standard uptake value (SUV_mean_) for ^18^FDG and ^125^I-BMIPP significantly correlated in these depots, although uptake of ^125^I-BMIPP in BAT and iWAT more closely mimicked the fold-change in metabolic rate as measured by an extracellular flux analyzer. Herein, we find that imaging BAT with the radioiodinated fatty acid analogue BMIPP yields more physiologically relevant data than ^18^FDG-PET/CT, and its conventional use may be a pivotal tool for evaluating BAT in both mice and humans.

## 1. Introduction

Combating obesity and obesity-related metabolic complications like diabetes mellitus, insulin resistance, or hyperlipidemia is a priority in the developed world. Before the COVID-19 pandemic, obesity was estimated to contribute to over 175,000 excess deaths per year in the United States [1]. Now, obesity is a known significant risk factor for COVID-19 mortality, particularly in younger patients [2,3,4]. Additionally, reducing obesity prevalence in the United States could lower healthcare spending [5]. Brown adipose tissue (BAT) is a mitochondrial rich and energetically demanding tissue responsible for thermogenesis and its presence has been linked to good metabolic health [6,7], making BAT a promising target for obesity-related diseases.

Histological analysis of human cadaver tissues by Juliet Heaton in 1972 revealed numerous subcutaneous and visceral depots of metabolically active fat tissue dispersed throughout the body [8]. Previously, metabolically active fat (commonly referred to as brown adipose tissue, or BAT) was thought to largely disappear after infancy in humans [9]. Heaton’s study, however, identified BAT around the deep organs at all life stages, ranging from infancy to late adulthood, as well as a wide distribution of several brown depots that recede or “whiten” with age.

The widespread use of ^18^FDG-PET/CT in cancer screening in the late 1990s and early 2000s renewed interest in BAT as the tissue was broadly identified across adult human populations [10,11]. Work began in earnest to characterize and manipulate BAT, and activation of this tissue by environmental or chemical stimulation has since been shown to be inversely correlated with BMI, insulin resistance, and serum triglycerides levels in humans and reduce body weight in mice [6,7]. Human subject studies often include an ^18^FDG-PET/CT scan to assess BAT activity [12,13]. At the time of writing, there are over 30 registered studies on Clinicaltrials.gov that include an ^18^FDG-PET/CT scan of BAT. Yet many known BAT depots are not reliably appreciated on ^18^F-FDG-PET/CT images, potentially limiting the utility of the method in fully defining the tissue. Surprisingly, ^18^FDG-PET/CT has never been validated against a tissue-based gold standard assay of metabolic activity despite physiologic reasons to believe that FDG uptake may not always accurately reflect BAT activity. The primary energy source of BAT is intracellular fatty acids, and glucose uptake is a downstream effect of BAT energy utilization [14]. Because BAT preferentially consumes fatty acid energy stores, a fatty acid tracer would likely be more mechanistically relevant for in vivo assessment than ^18^FDG.

Radioiodinated β-Methyl-p-iodophenyl-pentadecanoic acid (BMIPP) is a clinically applied radiolabeled branching long-chain fatty acid analogue that is typically used to appraise myocardial damage after an ischemic event [15]. Minimal β-oxidation of BMIPP traps the molecule in the intracellular triglyceride pool [16]. The slow metabolism of BMIPP as it accumulates in the cell makes it an excellent candidate for imaging BAT via single photon emission computed tomography (SPECT).

In addition to in vivo evaluation of classical mitochondria-dense BAT, our group has observed that ^125^I-BMIPP SPECT/CT also confers an advantage in visualizing white adipose that demonstrates a propensity for browning [17]. This non-classical BAT may be induced to exhibit a brown-like phenotype under cold exposure or chemical stimulation. Preclinical studies have long used the β_3_-adrenergic receptor agonist CL-316,243 to induce lipolysis in rodent models [18]. Depletion of triglycerides in adipose tissue enhances CD36 mediated transport of lipophilic molecules into brown adipocytes to replenish fatty acid energy stores [19]. Indeed, we have used CL-316,243 stimulation to exploit this mechanism and inspire greater BMIPP uptake in stimulated fat tissue as compared to cells or tissues that do not heavily utilize β-oxidation in their metabolic processes [17].

In this study, we directly compare ^18^FDG-PET/CT and ^125^I-BMIPP SPECT/CT imaging of adipose tissue in a CL-316,243 treated murine model against cellular bioenergetics. We hypothesized that while ^18^FDG and BMIPP uptake would be correlated in classical BAT, BMIPP imaging would be more sensitive than FDG for the detection of nonclassical BAT.

## 2. Results

### 2.1. Morphometric Data

Body mass did not differ between groups before receiving either CL-316,243 or saline vehicle injections (24.4 ± 0.3 g [treatment] vs. 24.4 ± 2.3 g [control], *p* = 0.7) or after (22.6 ± 2.3 g [treatment] vs. 23.3 ± 0.3 g [control], *p* = 0.3) the study period. However, percent body fat as measured by dual energy x-ray absorptiometry was decreased in treated mice following eight days of daily CL-316,243 injection (12.9 ± 1.9% vs. 18.9 ± 0.8%, *p* = 0.03), but not before (15.5 ± 3.0% vs. 17.1 ± 1.1%, *p* = 0.55).

#### 2.1.1. Imaging Metrics

^18^FDG mean SUV_mean_ was greater in the interscapular, inguinal, and gonadal fat depots of CL-316,243 treated mice than in vehicle control animals. ^125^I-BMIPP SUV_mean_ was greater in interscapular and inguinal fat pads in CL-316,243 mice compared to controls. Detailed results are shown in Table 1 and representative scans in Figure 1. Within untreated animals, there was a significant difference in both FDG (*p* = 0.0001) and BMIPP (*p* = 0.0001) uptake between fat depots, with greater uptake in BAT than in iWAT or gWAT, but no substantial difference between iWAT and gWAT SUV_mean_. A similar pattern was observed in CL-316,243 treated animals (*p* < 0.0001 for FDG and *p* = 0.006 for BMIPP). SUV_mean_ for FDG and BMIPP correlated in classical BAT (*r* = 0.82, *p* = 0.01) and inguinal fat (*r* = 0.80, *p* = 0.01), but not in gonadal fat (*r* = 0.27, *p* = 0.45). We also visually inspected uptake in other fat depots and noted a similar number of animals with both FDG and BMIPP uptake in the perivertebral and perirenal fat pads. Three mice had anterior abdominal subcutaneous BMIPP uptake compared to one mouse with FDG uptake in the same region. Two mice had periesophageal BMIPP uptake and no mice had FDG uptake in that area (qualitative data not shown). Quantitative autoradiography confirmed greater ^125^I-BMIPP uptake in CL-316,243 treated animals compared to controls in interscapular BAT (*p* = 0.04) and trends towards greater uptake in inguinal (*p* = 0.25) and gonadal (*p* = 0.22) fat (Figure 2).

#### 2.1.2. Metabolic Assay

The metabolic rate was significantly increased with CL-316,243 in all fat pads (Figure 3) as determined by Seahorse flux analyzer. Within each treatment group, interscapular BAT had a higher basal metabolic rate than iWAT (*p* = 0.001) and gWAT (*p* = 0.02). There was no significant difference between the metabolic rates of iWAT and gWAT.

#### 2.1.3. Tissue Studies

Histologic sections confirmed the type and morphology of adipose tissues used for metabolic assays (Figure 4). gWAT and iWAT depots in control animals were overwhelmingly white adipose tissue with very few multilocular fat droplets in adipocytes. In treated animals, iWAT and gWAT depots had interspersed multiloculated adipocytes. BAT depots in control animals had larger appearing fat droplets compared to treated animals. Immunohistochemistry with anti-TOMM40, a mitochondrial outer membrane protein, showed increased mitochondrial density in WAT with CL-316,243 treatment (Appendix A). Staining with anti-UCP1 in WAT showed almost no UCP1 content in control animals in contrast to robust presence of UCP1 in treated animals (Appendix A). The difference in mitochondrial density and UCP1 content between treatment and control was less pronounced in BAT compared to WAT (Appendix A).

*Ucp1* mRNA content increased in treated animals compared to controls in all fat depots. The largest fold increase was in gWAT (770-fold; *p <* 0.0001), followed by iWAT (19-fold; *p* < 0.01), and BAT (2.0-fold; *p* = 0.01). gWAT *Ucp1* absolute mRNA content in treated animals was lower than in untreated iWAT or BAT depots (Figure 5).

## 3. Discussion

The present study demonstrates that ^125^I-BMIPP and ^18^FDG uptake increase in BAT and WAT after treatment with the β_3_-agonist CL-316,243 in a manner consistent with the tissues’ metabolic activity as measured by an extracellular flux analyzer. Oxygen consumption increased 4-fold in iWAT and 1.6-fold in BAT following CL-316,243 treatment. ^125^I-BMIPP imaging closely resembled this pattern with a CL-316,243 induced three-fold increase and a 1.8-fold increase in iWAT and BAT uptake, respectively. ^18^FDG uptake was increased two-fold in iWAT by CL-316,243 and 2.8-fold in BAT. ^18^FDG SUV_mean_ was also significantly greater in the gWAT of CL-316,243 treated animals than in iWAT, a surprising finding as gWAT is not considered an inducible fat depot through sympathetic innervation. There was no change in ^125^I-BMIPP gWAT SUV_mean_. These findings suggest that ^125^I-BMIPP may provide more specific information about thermogenic metabolic activity in adipose than ^18^FDG.

^18^FDG-PET/CT became the de facto gold standard for imaging BAT because of its widespread clinical use in oncological applications. An NIH working group released recommendations for its use and reporting in human trials [20], including application of the term “brown adipose tissue” to any region of fat with detectable UCP1-positive adipocytes, encompassing inducible BAT. A review of large oncology cohorts generated important insights into the presence of BAT and epidemiological associations in humans. Reports in the early 2000s by clinical Nuclear Medicine physicians cautioning against mistaking supraclavicular BAT FDG uptake as nodal metastases were among the earliest indications that BAT is present in a significant number of adult humans [10,11]. Subsequent retrospective studies revealed greater BAT prevalence in younger, female, and leaner adults [21,22,23]. Increased BAT prevalence in the cold winter months in adult humans has also been noted through a review of clinical ^18^FDG-PET/CT scans [21,24]. Repeat scanning of the same human subjects shows variable BAT prevalence ranging from 20–100% [25,26,27]. Perhaps the greatest contribution of these findings is the recognition that BAT is present in a high number of adults and, when paired with knowledge regarding BAT’s association with metabolic health in small animals, may therefore be a targetable tissue for anti-obesity and metabolic pharmaceuticals. However, the mechanisms that are necessary and sufficient for FDG retention above background levels remain poorly understood. Even in humans, investigators use the term “activated BAT” to connote the fact that tracer uptake is most often not taken up over background tissues. When it is seen, most often it is in the supraclavicular region, while other depots, such as a perirenal, retroperitoneal, mediastinal, and paravertebral, infrequently display above-background FDG retention. Findings from retrospective studies do not confirm that ^18^FDG-PET/CT is the best imaging method for monitoring BAT activity in research settings where a wider selection of modalities are available.

In fact, there is empirical reason to believe ^18^FDG-PET/CT misses a large portion of anatomical BAT in humans since repeat scanning increases the rate of detection. Theoretically, BAT ^18^FDG uptake is thought to occur as metabolically active adipocytes replenish their lipid pools in part with glucose [14]. Different radiotracers have been used to assess BAT activity and presence with varying levels of success. Dynamic ^11^C-acetate imaging identifies tissues that convert the injected tracer into CO_2_ through the citric acid cycle at high rates and subsequently clear the ^11^C as carbon dioxide. Dedicated centers with onsite cyclotrons have used ^11^C-acetate to identify and monitor BAT activity [14,28]. Be that as it may, widespread use of ^11^C-acetate in this setting is limited by the need for specialized equipment and technical expertise. Like ^18^FDG, ^11^C-acetate also will not detect quiescent BAT. Another radiotracer strategy is to measure fatty acid uptake. Labbé and colleagues showed that thermogenic adipocytes primarily use the oxidation of intracellular fatty acids to fuel energy expenditure [14] and thermogenic adipocytes take up free fatty acids through CD36 [19,29]. The free fatty acid analogue ^18^F-fluoro-6-thia-heptadecanoic acid (FTHA) has shown greater sensitivity for nonclassical BAT depots in rodents [14] and attainability in human studies [28,30].

A handful of studies performed by us and others demonstrate that fatty acid-based imaging of thermogenic adipose depots with BMIPP is feasible [17,31,32]. In 2018, our group anatomically mapped the fat pads in a murine model using a combination of ^18^FDG-PET and ^125^I-BMIPP-SPECT imaging at basal conditions and following β_3_-adrenergic stimulation or cold exposure. In our hands, small animal imaging with ^123^I and ^125^I-BMIPP has greater signal-to-noise ratio and better image quality than FTHA and identifies nonclassical BAT depots in a manner similar to FTHA [17]. However, use of BMIPP in the field is atypical. While these studies found that BMIPP-SPECT may be advantageous for identification of brown and beiging adipose, a direct comparison between FDG-PET and BMIPP-SPECT has remained absent from the literature. This study builds on our previous qualitative work to definitively show that BMIPP offers superior visualization of classical and more importantly inducible BAT.

We associated ^18^FDG-PET and ^125^I-BMIPP-SPECT signal intensity in various adipose tissues with histological sections, autoradiography, and, for the first time, mitochondrial respiration. ^125^I-BMIPP uptake more accurately reflected the metabolic activity of BAT and iWAT than ^18^FDG, which measures Glut transporter and hexokinase activity as opposed to fatty acid oxidation or lipolysis. We found that while ^18^FDG and ^125^I-BMIPP uptake are correlated in classical BAT, ^125^I-BMIPP-SPECT/CT also identifies more regions of nonclassical metabolically active fat in periesophageal and subcutaneous adipose tissue. We were surprised to find a marked increase in gWAT metabolic activity with treatment as this fat depot is often considered to have little potential for inducible browning. *Ucp1* mRNA content and UCP1 immunohistochemistry staining also increased markedly in gWAT, indicating a true increase in the BAT phenotype. We believe this unexpected result may be due to a combination of our use of BALB/c mice, which are phenotypically leaner and typically metabolically healthier than the C57BL/6 strain [33], and the delivery of CL-316,243 by intraperitoneal injection that bathed the peritoneum with the drug. A similar phenomenon has been seen in humans with subdiaphragmatic paragangliomas [34]. ^18^FDG but not ^125^I-BMIPP uptake increased with treatment in gWAT, which may indicate that a different set of physiologic changes occur when adipose tissue is exposed to local versus systemic β_3_ agonism.

To date, no molecular imaging technique for quantitatively measuring BAT activity has been directly compared to a tissue-based gold-standard assay of metabolism. Our study identifies BMIPP as a relevant candidate for such a study. Based on our findings herein, we believe the field is not fully realizing the potential of BMIPP as a powerful BAT imaging tool. Both ^18^FDG and BMIPP share certain disadvantages, such as radiation exposure, cost, and low spatial resolution, although as modern scanners reach spatial resolutions of 7 mm or smaller, this has become a less limiting factor. The most important element in nuclear imaging remains the amount of tracer that is retained in the target tissue relative to surrounding background tissue. ^18^FDG uptake in fat depots is notoriously inconsistent in human subjects. Commonly advanced predictive factors for FDG accumulation in fat, such as ambient temperature, body mass, or female gender are not decisively reproducible, as the senior author has seen in his own clinical practice and is evidenced by serial scanning in humans [25,26,27]. To compensate for this shortcoming, investigators have invoked such terms as “activated brown fat” or “metabolically active fat”. Regardless, ^18^FDG-PET/CT identification of brown or browning fat pads is demonstrably variable.

^18^FDG-PET is not the only modality that has been leveraged for BAT imaging. Magnetic resonance imaging, magnetic resonance spectroscopy, near-infrared spectroscopy, infrared thermal imaging, and ultrasound are spatially limited and disproportionately concentrate on the supraclavicular fat pad [35,36,37,38]. Supraclavicular BAT tends to be most conspicuous on ^18^F-FDG-PET imaging, but the focus on head and neck BAT is an incomplete picture of BAT depots. In humans, metabolically active adipocytes are interspersed within white adipose depots [39,40], a pattern more like murine inducible thermogenic adipose tissue than classical interscapular BAT. Considering the distribution of metabolically active fat throughout the human body, whole body imaging is important to conduct a full and thorough investigation of the tissue. Therefore, there is an unmet need for a radiotracer that can sensitively image brown or beiging fat for measuring the impact of fat burning medications, weight loss regimens, or environmental conditions. We believe that SPECT/CT imaging with BMIPP has the potential to meet this need as a whole-body method capable of identifying more comprehensively those fat depots that are utilizing the intracellular fat stores for energy or heat production. There are now clinically available digital SPECT scanners that make whole body SPECT a practical reality (GE StarGuide, Spectrum Dynamics VERITON). BMIPP is already used clinically for the assessment of myocardial fatty acid metabolism but implementing BMIPP-SPECT for whole body fat imaging could enhance further investigation into the mechanistic relationship between various BAT depots and obesity. Moreover, ^124^I could be substituted for ^125^I in clinical imaging to take advantage of the higher sensitivity and spatial resolution of clinical PET scanners compared with SPECT scanners, and could enable dual modality PET/CT or PET/MRI studies.

Our study has limitations. First, the ex vivo and in vitro studies were not performed on the same animals as imaged due to the long 60-day half-life of ^125^I. However, we ran a parallel study with the same treatment exposure to demonstrate the physiologic changes underpinning our imaging results. Additionally, the study is somewhat limited by sample size. Nonetheless, the treatment protocol we used created differences in metabolic activity that are significant even with this relatively small number of animals.

The advantage of BMIPP over FDG is clear in that FDG remains mostly reliable for imaging classical BAT in rodents. We show that BMIPP is better suited for imaging inducible browning or beige adipose tissue compared with FDG. Further work will allow researchers to provide a more sensitive and informative imaging assay of BAT activity to metabolic researchers.

## 4. Materials and Methods

### 4.1. Animal Protocol

All animal studies were approved and conducted under the oversight of the University of Texas Southwestern Institutional Animal Care and Use Committee. The University of Texas Southwestern uses the Guide for the Care and Use of Laboratory Animals of the National Institutes of Health when establishing animal research standards.

Male BALB/c mice aged eight weeks were purchased from UTSW’s internal breeding core facility. Littermates were randomly assigned to CL-316,243 treatment or saline vehicle control groups. Five animals were used for both groups in all studies, except where otherwise indicated. Animals were treated with either 1 mg/kg CL-316,243 dissolved in 100 μL normal saline or an equivalent volume of the saline vehicle via intraperitoneal injection daily for eight days. Mice were housed at ambient temperature with 12-h light/dark cycles and ad libitum access to a normal chow diet (Envigo Teklad 2016). Animals were sacrificed according to institutional approved protocol at the end of the study period. CL-316,243 treated and vehicle control pairs underwent molecular imaging with ^18^FDG-PET/CT and BMIPP-SPECT/CT on consecutive days at the end of the study period (Days seven and eight). A total of 10 mice (*n* = 5 per group) were used for all imaging studies. The imaging study design is illustrated in Figure 6.

Additionally, a parallel set of CL-316,314 treated and vehicle control animals (*n* = 5 per group) had their adipose tissue basal metabolic rate assessed using a Seahorse Xfe24 extracellular flux analyzer (Agilent, Santa Clara, CA, USA) on Day eight. gWAT, iWAT, and BAT were harvested from all animals for tissue studies. Fat depots used for immunohistochemistry or histology were immediately fixed in 10% neutral buffered formalin and tissue used for qPCR was snap frozen in liquid nitrogen and stored at −80 °C. Due to the long 60-day half-life of ^125^I, we could not perform tissue studies on mice following ^125^I-BMIPP injection.

### 4.2. Radiotracers

^18^F-FDG was purchased from PETNET (Dallas, TX, USA). ^125^I-BMIPP was prepared by the method previously described by Goodman [41].

### 4.3. ^18^F-FDG-PET/CT

Mice fasted overnight prior to ^18^FDG-PET/CT on an Inveon PET/CT (Siemens Molecular Imaging, Knoxville, TN, USA). On the day of scanning, mice were briefly anesthetized with isoflurane prior to injection of 3.7 MBq ^18^F-FDG via the lateral tail vein. They were allowed to roam in a cage during a one-hour uptake period. They were re-anesthetized and maintained under isoflurane anesthesia throughout the PET/CT. Whole body CT images were obtained at 80 kV with a focal spot of 58 μm and a binning factor of 1:4. A whole body PET was then obtained.

### 4.4. ^125^I-BMIPP-SPECT/CT

Thyroid iodine uptake was suppressed by supplementing drinking water with 1:1000 Lugol’s solution (10 g KI + 5 g I_2_ dissolved in 100 mL deionized water) for three days prior to scanning. Mice were fasted overnight before ^125^I-BMIPP-SPECT/CT. On the day of the SPECT/CT, mice were briefly anesthetized with isoflurane and injected with ~17.5 MBq ^125^I-BMIPP via the lateral tail vein. They were allowed to roam in a cage during a one-hour uptake period. After a one-hour uptake period, mice were re-anesthetized and kept under isoflurane anesthesia during a SPECT/CT on a NanoSPECT/CT Plus System (Mediso, Budapest, Hungary). After acquiring a localizer topogram, SPECT data was acquired with a four-detector configuration using 9-pinhole aperture plates with a resolution of 0.74 mm. A CT was then performed using 360 projections per rotation with 45 kVp, 1000 ms exposure, and a binning factor of 1:4.

### 4.5. Image Analysis

VivoQuant (software version 2.0, Invicro, Needham, MA, USA) and Inveon Research Workplace (software version 4.2, Siemens Medical Solutions, Knoxville, TN, USA) were used to co-register CT with SPECT and PET images, respectively, and perform image analysis. Regions of interest (ROIs) were drawn in interscapular, inguinal, and gonadal fat depots, as these are the primary sites of classical baseline BAT activity, β3-induced BAT activity, and poorly inducible BAT, respectively [17,42]. Classical interscapular BAT depots were manually contoured using BMIPP and FDG uptake and CT density as a guide. A threshold of HU −180 to −10 was applied to distinguish adipose tissue from muscle uptake [20]. Two small circular regions of interest were drawn in the inguinal fat depots immediately superior to the hip on anterior-posterior images using a regional lymph node as a guide [17]. Two small circular ROIs were drawn 45 degrees anterolateral to the superior bladder to each side in the gonadal fat pad. Examples of ROIs are in Figure 7. Mean and max standardized uptake values (SUV) were calculated in each ROI for BMIPP-SPECT/CT and ^18^FDG-PET/CT.

### 4.6. Autoradiography

Freshly dissected fat pads were placed in an optimal cutting temperature compound (Sakura Finetek USA, Torrance, CA, USA) and frozen overnight at −20 °C. A Leica (Buffalo Grove, IL) CM1950 cryostat was used to cut 20 μm slices of tissue that were placed onto glass slides. Slides were then placed on a storage phosphor screen for 2 h before image acquisition with a Cyclone Plus storage phosphor system (Perkin Elmer, Waltham, MA, USA). ROIs were drawn around adipose tissue sections in OptiQuant 5.0.0.2 (Perkin Elmer) and the density light unit (DLU)/mm^2^ obtained for each fat pad.

### 4.7. DEXA

A body composition analysis was performed with whole-body densitometry using a PIXImus densitometer (GE Lunar Corporation, Madison, WI, USA). Lunar PIXImus software version 2.10.041 was used for analysis. Animals were anesthetized with isoflurane before and during image acquisition. The densitometer was calibrated with a manufacturer-provided aluminum/Lucite phantom before each use. Total body mass, lean tissue mass and percent body fat were calculated for an ROI excluding the head, as is standard practice [43].

### 4.8. Metabolic Assay

Mitochondrial oxygen consumption rates (OCRs) from dissected adipose tissue depots were determined with a Seahorse XF24 Extracellular Flux Analyzer (Agilent, Santa Clara, CA, USA). Due to the low volume of the classical BAT depots, the entire left BAT pad was utilized for this assay. The right classical BAT depot was reserved for further tissue studies. Sections of the inguinal and gonadal fat depots corresponding to imaging ROIs were used. Approximately 2–4 mg interscapular BAT or 8–10 mg inguinal or gonadal adipose tissue was placed in each assay well. The assay was performed as previously described [44].

### 4.9. Histology and Immunohistochemistry

Fat pads were fixed, embedded, and stained for immunohistochemistry as previously described [44]. Rabbit polyclonal antibody to UCP1 (ab10983, Abcam, Cambridge, MA, USA) and TOMM40 (ab272921, Abcam) were used.

### 4.10. qPCR

mRNA was isolated from homogenized fat depots using TRIzol reagent (Invitrogen, Carlsbad, CA, USA). cDNA was then synthesized by reverse transcription of RNA (High-Capacity cDNA Reverse Transcription Kit, Applied Biosystems, Foster City, CA, USA) and gene expression evaluated by quantitative real-time PCR. Commercial TaqMan Universal PCR Master Mix and TaqMan Gene Expression Assay kits (Applied Biosystems) were used to determine mRNA expression levels of *Ucp1* (Mm01244861_m1), *Pparg2* (Mm00440940_m1), *Prdm16* (Mm00712556_m1), *Dio2* (Mm00515664_m1), and *Srebf1* (Mm00550338_m1). *Hprt* (Mm03024075_m1) was used as a housekeeping gene for each assay. Tissues were prepared and analyzed as previously described [44]. Analysis could not be completed on samples from one CL-316,243-treated mouse due to technical failure.

### 4.11. Statistics

Values are reported as mean ± standard deviation. All data met assumptions of normality based on distribution histogram analysis. Differences between groups were compared with *t*-tests. Correlations between continuous variables were assessed with the Pearson correlation coefficient. All analyses were performed with SAS version 9.4 (SAS, Cary, NC, USA).

## 5. Conclusions

BMIPP is a promising radiotracer for quantitative BAT imaging. A logical next step to assess this radiotracer’s utility for this purpose is to compare imaging metrics with ^123^I-BMIPP directly to the tissue-based metabolism, which is possible because of ^123^I’s short half-life. Moreover, ^123^I is amenable to clinical imaging. An ideal imaging technique will detect BAT interspersed in WAT with high sensitivity and provide reproducible and interpretable quantitative data. Given the recent demonstration that BAT activity is an indicator of better cardiovascular health and insulin sensitivity [45], non-invasive assays that measure its activity and location may be important for determining incidence/prevalence in a population and development of therapies.

## Figures and Tables

**Figure 1 ijms-23-04880-f001:**
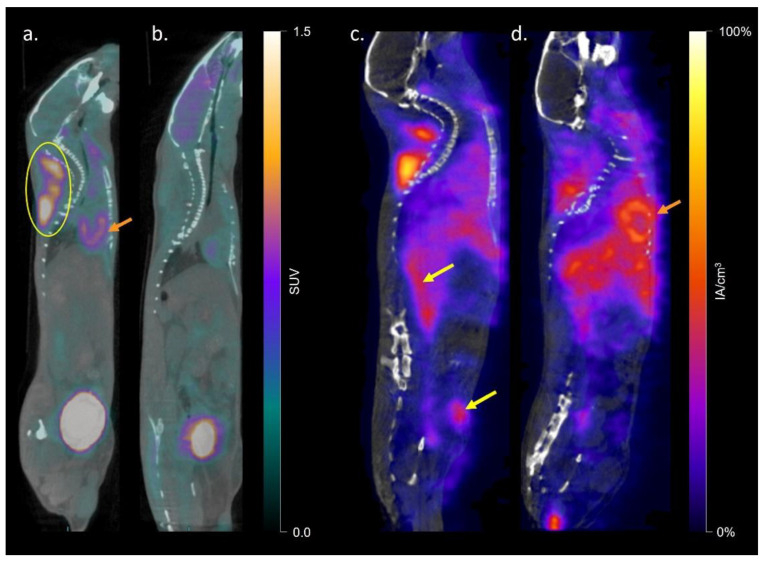
Representative sagittal ^18^FDG-PET/CT of a CL-316,243 treated (**a**) and vehicle control (**b**) mouse and ^125^I-BMIPP-SPECT/CT of a CL-316,243 (**c**) and vehicle (**d**) mouse (*n* = 5). The classical interscapular fat pad is circled in image (**a**), cardiac uptake is indicated with orange arrows in (**a**,**d**), and prevertebral and anterior abdominal fat uptake indicated with yellow arrows in (**c**). Extensive hepatic, gastrointestinal, and mediastinal background uptake was seen with BMIPP but not FDG.

**Figure 2 ijms-23-04880-f002:**
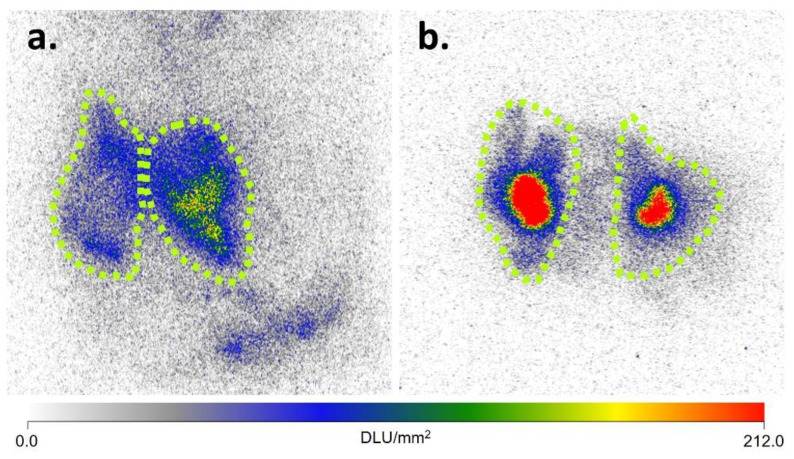
Autoradiography of ^125^I-BMIPP in bilateral BAT depots of vehicle treated control (**a**) and CL-316,243 treated (**b**) mice. The area of each BAT pad is delineated by the dashed perimeters. Scalebar is represented in digital light units (DLU)/mm^2^.

**Figure 3 ijms-23-04880-f003:**
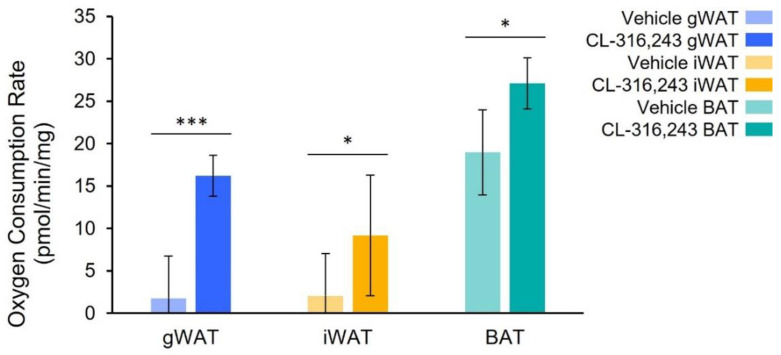
Basal metabolic rate from selected fat pads in CL-316,243 treated and vehicle mice. CL-316,243 treatment increased the metabolic rate significantly in all fat pads, *n* = 5 for all groups (* indicates *p* < 0.05, *** *p* < 0.001).

**Figure 4 ijms-23-04880-f004:**
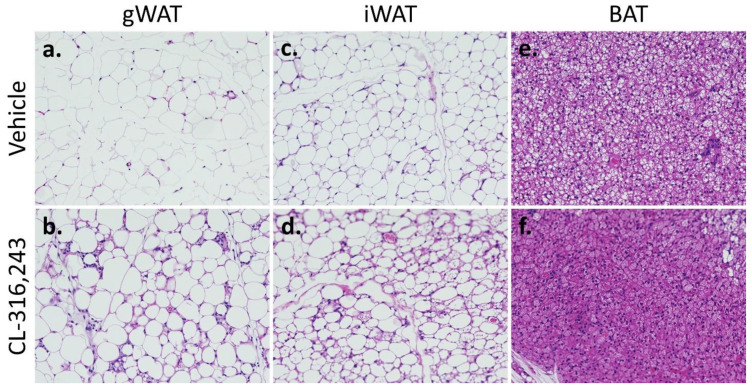
Histology demonstrates increased brown phenotype in mice treated with CL-316,243. 20× Representative H&E stained sections from control (**a**) and treated (**b**) gWAT, control (**c**) and treated (**d**) iWAT, and control (**e**) and treated (**f**) BAT. Note the increased multilocular adipocytes in treated gWAT and iWAT compared to control and decreased lipid droplet size in treated versus control BAT.

**Figure 5 ijms-23-04880-f005:**
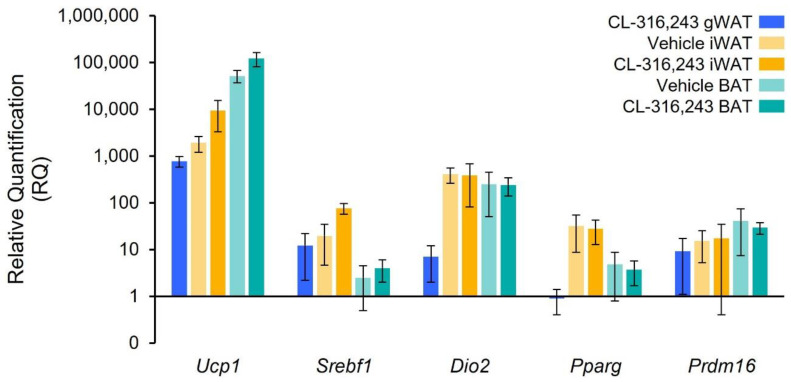
Relative quantification (RQ) of assayed genes. Values were normalized to mRNA expression in gWAT of vehicle mice. Data are reported as the mean expression value with 95% CI in vehicle (*n* = 5) and CL-316,243 treated mice (*n* = 4). RQ is shown on a logarithmic scale.

**Figure 6 ijms-23-04880-f006:**
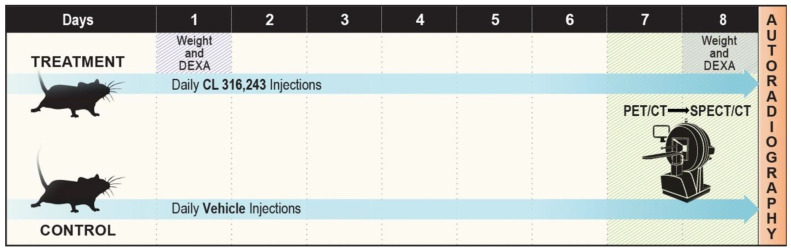
Imaging study protocol schematic. Mice were randomized to treatment with CL-316,243 or vehicle control injections with normal saline. Dual energy X-ray absorptiometry (DEXA) was performed on Days one and eight to assess body fat composition. ^18^FDG-PET/CT was performed on Day seven and BMIPP-SPECT/CT on Day eight. Animals were sacrificed after SPECT/CT for autoradiography.

**Figure 7 ijms-23-04880-f007:**
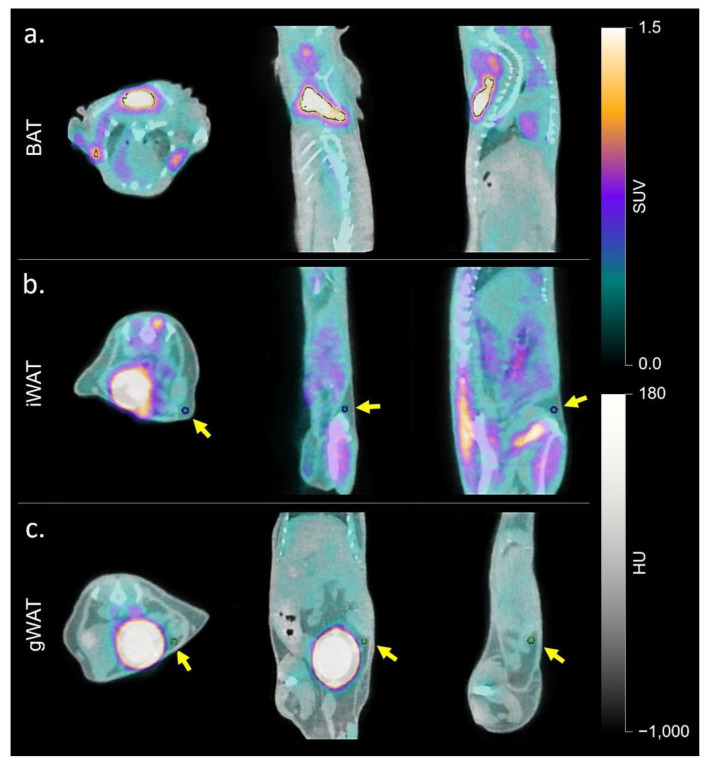
Examples of ROIs on an ^18^FDG-PET/CT in (**a**) classical interscapular BAT, (**b**) inguinal WAT, and (**c**) gonadal WAT. Yellow arrows indicate spherical ROIs positioned within the iWAT and gWAT depots in (**b**,**c**).

**Table 1 ijms-23-04880-t001:** Mean FDG and BMIPP SUVs in selected adipose tissue depots *.

Fat Pad	FDG SUV_mean_		*p*	BMIPP SUV_mean_		*p*
	CL-316,243	Vehicle		CL-316,243	Vehicle	
Interscapular	1.88 ± 0.64	0.68 ± 0.25	0.008	5.95 ± 1.44	3.34 ± 1.17	0.03
Inguinal	0.196 ± 0.035	0.096 ± 0.050	0.01	1.99 ± 0.77	0.67 ± 0.44	0.01
Gonadal	0.268 ± 0.103	0.092 ± 0.062	0.02	0.608 ± 0.208	0.429 ± 0.140	0.19

* Means compared with *t*-tests, *n* = 5.

## Data Availability

Not applicable.

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
