# Peer review of "Comparison of BMIPP-SPECT/CT to 18FDG-PET/CT for Imaging Brown or Browning Fat in a Preclinical Model"

_ijms, 2022, doi:10.3390/ijms23094880_

Round 1

Reviewer 1 Report

The authors have demonstrated that instead of FDG-PET/CT, BMIPP-SPECT/CT can be used to evaluate the BAT activity in mice. I agree with the data, but I am not sure regarding the novelty of the study. Your group and another one have already shown that iBAT can be evaluated using BMIPP-SPECT/CT. Moreover, I wonder what the authors want for advancing to clinical practice. The disadvantage of 18FDG-PET/CT is radio exposure, cost, and special resolution. Human studies have attempted to address these problems of BAT activity measurement using MRI, NIRS, IRT, and US. Unfortunately, this study did not present the advantages of BMIPP-SPECT/CT over 18FDG-PET/CT. I am willing to accept it in another paper; however, I think it is still challenging in this high-impact Journal (IF = 4.6).

1) What is the novelty of this study? I am not sure regarding the difference between this study and your previous one (Cell Met 2018).  Furthermore, another group has already used BMIPP-SPECT/CT to evaluate BAT since 2014 (doi: 10.1016/j.bbrc.2014.12.124.) (10.1371/journal.pone.0090825).

2) Line 61: Please clarify what do the authors mean by “classical” and “nonclassical”. I understand these words, but they appeared without any introduction.

3) Lines 22 and 56: I think fatty acids from WAT are more important than intracellular fatty acids. I believe the authors should use the word “fatty acid” instead of “lipids” since you have measured fatty acid uptake.

4) Please change the term FDG-PET/CT or FDG-PET to 18FDG-PET/CT.

5) I wonder if you can show the correlation graph between 18FDG-PET/CT and BMIPP-SPECT/CT, mentioned in the abstract and objective.

6) Table 1: Please explain the meanings of the treated and control groups for better clarity. I understand that you have written it in the methods and Fig. 5, but aTable should be self-explanatory.

7) Fig. 4: Please change the term to Relative “Quantification” instead of “Expression”.

8) Lines 174–176: The authors should write how much is the difference to help make this sentence objective?

9) Line 188: BAT

Author Response

Please see attached Word document.

Reviewer 2 Report

In this work, Frankl and collaborators display an interesting work about the use of fatty acid tracer instead of classic glucose tracer to analyze brown and beige adipocyte activity. This is a well-conducted study, well-written and leading to an interesting discussion. This work corresponds to a pilot study demonstrating the importance of BMIPP tracer in metabolism analysis. I’ve some minor comments. Several are mandatory and the others can improve the clarity of this paper.

  • BMIPP must be described and its interest explained at the beginning of the paper (not only in the abstract).
  • It is unclear what mechanism the authors want to reveal with a FA tracer. Indeed, the authors claimed with accuracy that thermogenic adipocytes use preferentially their store of fatty acids, and that glucose and FFA are used to refill Triglycerides stores. For my point of view, acute BMIPP injection at the end of CL316,243 treatment allows evaluation of FFA uptake by thermogenic adipocyte (and other cells able to do this). What is the point of view of the authors? This point must be introduced and discussed.
  • CL316,243 induces lipolysis. Is this liberation of FFA in circulation can limit/alter BMIPP analysis?
  • The numbers of mice/fat pad/tissue for each analysis and groups are never indicated. These must be added clearly in all figure and table legends and in material and methods.
  • What are the results displayed in figure 4? It is not a mean +/- sem.
  • What is the isoform of PPARg analyzed by qPCR? Only PPARg2 has an interest to evaluate browning.
  • t-test instead of T-test

Author Response

Please see attached Word document.

Round 2

Reviewer 1 Report

I acknowledge your polite answer. Great Job!
Thank you so much.